# COVID-19 vaccination beliefs, attitudes, and behaviours among health and social care workers in the UK: A mixed-methods study

**Sadie Bell**[1☯]*, **Richard M. Clarke**[2☯], **Sharif A. Ismail**[1], **Oyinkansola Ojo-Aromokudu**[1], **Habib Naqvi**[3], **Yvonne Coghill**[3], **Helen Donovan**[4], **Louise Letley**[5], **Pauline Paterson**[6‡], **Sandra Mounier-Jack**[1‡]

1 Department of Global Health and Development, Faculty of Public Health and Policy, London School of Hygiene & Tropical Medicine, London, United Kingdom, 2 University of Southampton, Southampton, United Kingdom, 3 NHS Race and Health Observatory, London, United Kingdom, 4 Royal College of Nursing, London, United Kingdom, 5 Immunisation, Hepatitis & Blood Safety Department, National Infection Service, Public Health England, London, United Kingdom, 6 Department of Infectious Disease Epidemiology, Faculty of Epidemiology and Population Health, London School of Hygiene & Tropical Medicine, London, United Kingdom

☯ These authors contributed equally to this work.
‡ PP and SMJ also contributed equally to this work.
* sadie.bell@lshtm.ac.uk

**Data Availability Statement:** The full quantitative survey dataset is stored in a data repository: LSHTM Data compass. Data are available to bona

## Abstract

### Background

The UK began delivering its COVID-19 vaccination programme on 8 December 2020, with health and social care workers (H&SCWs) given high priority for vaccination. Despite well-documented occupational exposure risks, however, there is evidence of lower uptake among some H&SCW groups.

### Methods

We used a mixed-methods approach—involving an online cross-sectional survey and semi-structured interviews–to gain insight into COVID-19 vaccination beliefs, attitudes, and behaviours amongst H&SCWs in the UK by socio-demographic and employment variables. 1917 people were surveyed– 1656 healthcare workers (HCWs) and 261 social care workers (SCWs). Twenty participants were interviewed.

### Findings

Workplace factors contributed to vaccination access and uptake. SCWs were more likely to not be offered COVID-19 vaccination than HCWs (OR:1.453, 95%CI: 1.244–1.696). SCWs specifically reported uncertainties around how to access COVID-19 vaccination. Participants who indicated stronger agreement with the statement *'I would recommend my organisation as a place to work'* were more likely to have been offered COVID-19 vaccination (OR:1.285, 95%CI: 1.056–1.563). Those who agreed more strongly with the statement *'I feel/felt under pressure from my employer to get a COVID-19 vaccine'* were more likely to have declined vaccination (OR:1.751, 95%CI: 1.271–2.413). Interviewees that experienced

fide researchers upon request and agreement by the study team. Data cannot be shared without restriction as study participants agreed that their study data could be accessed by other researchers only. This level of data access was approved by the LSTHM Observational Research Ethics committee (study reference: 22923). Although permission was granted by participants to share their interview and open-text survey data, there are concerns about potential harms that could come to participants given the content of this data. Therefore, we are sharing the anonymised quantitative survey data only. Further information on the data and access conditions can be found through the LSHTM Data Compass at: https://doi.org/10.17037/DATA.00002525. Researchers interested in accessing the data are advised to request access through the LSHTM Data Compass page listed above, email the corresponding author or email researchdatamanagement@lshtm.ac.uk.

**Funding:** The author(s) disclosed receipt of the following financial support for the research, authorship, and/or publication of this article: The research was funded by the National Institute for Health Research Health Protection Research Unit (NIHR HPRU) in Immunisation at the London School of Hygiene and Tropical Medicine (LSHTM) in partnership with Public Health England (PHE) (reference number NIHR200929); and by the NHS Race and Health Observatory. The views expressed are those of the author(s) and not necessarily those of the NHS, the NIHR, the Department of Health or Public Health England, or the NHS Race and Health Observatory. SAI is supported by a Wellcome Trust Clinical Research Training Fellowship [reference number 215654/Z/19/Z].

**Competing interests:** The authors have declared that no competing interests exist.

employer pressure to get vaccinated felt this exacerbated their vaccine concerns and increased distrust. In comparison to White British and White Irish participants, Black African and Mixed Black African participants were more likely to not be offered (OR:2.011, 95%CI: 1.026–3.943) and more likely to have declined COVID-19 vaccination (OR:5.550, 95%CI: 2.294–13.428). Reasons for declining vaccination among Black African participants included distrust in COVID-19 vaccination, healthcare providers, and policymakers.

## Conclusion

H&SCW employers are in a pivotal position to facilitate COVID-19 vaccination access, by ensuring staff are aware of how to get vaccinated and promoting a workplace environment in which vaccination decisions are informed and voluntary.

## Background

The UK began delivering its COVID-19 vaccination programme on 8 December 2020, prioritising allocation by age and to frontline health and social care workers (H&SCWs), as advised by the Joint Committee on Vaccination and Immunisation [1]. Frontline H&SCWs (i.e. those involved in direct service user care or who may have contact with service users) [2] were prioritised in the first phase of the vaccination programme due to their increased personal risk of COVID-19 exposure and of transmitting COVID-19 to vulnerable service users [1]. The UK healthcare workforce is made up of around 1.9 million people (68% in the publicly funded National Health Service (NHS), 32% in the private sector) [3] and social care employs approximately 1.52 million people [4].

As of 11 April 2021, COVID-19 vaccine uptake (first dose) in England was estimated at 86.2% in NHS Trust healthcare workers (HCWs) who appear in the NHS Electronic Staff Record and 70.3% in social care workers (SCWs) [5]–highlighting large differences in uptake between the sectors, which are not fully understood. Emerging data for England also indicates that COVID-19 vaccination uptake amongst H&SCWs varies by geographical area [5]. Patterns of seasonal influenza vaccine uptake in HCWs suggest that COVID-19 vaccination uptake may also vary markedly by workplace and job role [6].

People from certain ethnic minority backgrounds, including those in H&SC roles, are at elevated risk of contracting COVID-19 and at increased risk of adverse outcomes [7–10]. A disproportionately high number of Black and Asian H&SCWs have died during the COVID-19 pandemic, representing 21% of the NHS workforce but accounting for 63% of deaths in H&SCWs in the first wave of the pandemic (up until 22 April 2020) [11]. Despite this risk, early reports of COVID-19 vaccination uptake amongst HCWs have identified lower levels of uptake amongst certain ethnic minority groups [12], which mirror trends in the general population [13].

Our study aimed to identify and gain insight into factors influencing COVID-19 vaccination amongst H&SCWs in the UK, with a focus on understanding variations in vaccination beliefs, attitudes, and behaviours by socio-demographic factors, including ethnicity. In this paper, we use the term ethnic minority to refer to all ethnic groups except the White British & White Irish grouping.

## Methods

We used a mixed-methods approach involving a cross-sectional online survey, which included fixed and free-text questions, and follow-up semi-structured interviews. Ethical approval for this study was granted by the London School of Hygiene & Tropical Medicine Observational Research Ethics Committee (Reference: 22923).

### Cross-sectional survey

**Survey design and recruitment.** We performed an online survey of UK frontline H&SCWs between January 22nd and February 8th 2021. The survey was advertised through the communication streams of organisations with high levels of H&SCW membership/followers (e.g. via social media platforms, email, and staff newsletters). Recruitment aimed to reach H&SCWs from across job roles and ethnic minority groups by approaching organisations targeted at different occupational groups and ethnicities.

The survey (S1 Appendix) included four main sections, comprised of questions relating to:

1) Demographics: Age, gender, religion, long-term disability, country of birth and ethnicity. The ethnicity question was followed by a 5-item ethnic identity scale, the Multigroup Ethnic Identity Measure—Revised (MEIM-R) [14]. The MEIM-R scale gives an indication of the level to which individuals feel a close sense of belongingness and attachment towards their ethnicity.

2) Employment: Job sector, occupational group, role and grade; level of service user contact; job satisfaction, workplace discrimination (from colleagues or service users); experience of working in the pandemic (i.e. contact with service users at high risk of COVID-19 or diagnosed with COVID-19, and times asked to self-isolate); and the participant's COVID-19 status.

3) COVID-19 vaccination beliefs and COVID-19 risk perceptions: Beliefs around COVID-19 vaccine safety, effectiveness, importance, compatibility with participant's religion, and social norms related to COVID-19 vaccination were measured on 4-point Likert scales from (1) strongly disagree to (4) strongly agree; COVID-19 vaccination intentions (if not yet offered vaccination) and behaviours (decision to accept or decline vaccination); seasonal influenza vaccination history (for 2019/20 and 2020/21); and workplace pressure to accept COVID-19 vaccination.

4) Sources of COVID-19 vaccination information: Rating of trust in COVID-19 vaccination information sources were measured using the root statement '*I trust the advice on COVID-19 vaccination given by...*' followed by a range of information sources (e.g. the NHS, the government, friends and family members). Participants responded to each statement using a 5-point Likert scale from (1) strongly disagree to (5) strongly agree, with the mid-point labelled '*no opinion*'.

Free-text boxes were included for participants to provide their main reason(s) for accepting or declining COVID-19 vaccination and a text box was included at the end of the survey for further comments on views and experiences of COVID-19 vaccination.

We developed the survey in consultation with Public Health England, Royal College of Nursing and NHS Race and Health Observatory representatives. Several included questions were pre-existing survey questions that had already undergone testing (e.g. questions included in the demographics section). For new items and questions, face validity was gained through discussion with the various research stakeholders, and feedback on survey design (including the appropriateness and comprehensibility of questions) was obtained from a number of

H&SCWs. Factor analysis (see S2 Appendix) identified and confirmed underlying components of the *Vaccine belief* and *Trust in information source* items. Public Health England, Royal College of Nursing and NHS Race and Health Observatory representatives were involved in sharing the survey.

**Data recording and missing data.** Recruitment to the study led to 2307 survey link click-throughs. Of these, 388 cases were removed due to the participants not responding to questions beyond the 50% survey progress mark. Two further cases were removed due to ineligibility (i.e. participants were not frontline H&SCWs currently working in the UK), leaving 1917 included participants.

In order to run the logistic regressions with an appropriate number of participants in each of the subcategories of the categorical variables, the variables *Ethnicity* and *Job role* were recoded from 20 categories and 16 categories, respectively, into an *Ethnicity* variable with 7 categories and a *Job role* variable with 6 categories. In addition, we performed a factor analysis on the *Vaccine belief* and *Trust in information source* items to reduce the number of variables in the regression models. This reduced the 13 *Vaccine belief* items into two composite variables; *Combined COVID-19 Vaccine belief (important, safe, and effective)* (Cronbach's Alpha = 0.918) and *Social norms to vaccinate against COVID-19* (Cronbach's Alpha = 0.661) and 4 single items. The 12 *Trust in information source* items were reduced to three composite variables *Trust in health system sources* (Cronbach's Alpha = 0.876), *Trust in non-health system sources* (Cronbach's Alpha = 0.738), and *Trust in Friends and Family members* (Cronbach's Alpha = 0.876). Further details related to the combining of the ethnicity and job role categories and the factor analysis can be found in S2 Appendix.

Participants were not required to answer every question in the survey. Missing data were assessed to be low and missing at random. Multiple imputation was used to replace missing data in continuous variables for use in each logistic regression model and comparisons.

**Survey analysis.** We conducted a forward stepwise logistic regression to examine the association between demographic and employment variables, and seasonal influenza vaccination history, with being offered COVID-19 vaccination. To examine factors associated with vaccine uptake, we conducted a similar forward stepwise logistic regression to that of being offered vaccination, with the addition of the COVID-19 vaccine belief and trust in information sources variables. Finally, we assessed the differences across ethnicity with respect to COVID-19 related beliefs and trust in sources of information through a series of comparisons. Comparisons between White British & White Irish participants and each ethnic minority group were assessed through multiple ANOVA tests and Turkey post hoc analysis. All statistical analyses were conducted using SPSS V.27.

Free-text responses were analysed thematically in Microsoft Excel by SB, PP and SM-J. Coding schemes were produced based on the content of the free-text comments.

## Semi-structured interviews

**Recruitment.** On survey completion, participants were asked to provide their contact details if they were interested in taking part in a 30-45-minute follow-up interview. In selecting interested participants to interview, we aimed to maximise participant diversity in ethnicity, gender, job sector, occupational group, and geographical location. We prioritised contacting H&SCWs that had declined COVID-19 vaccination.

**Interview design and conduct.** Participants were emailed an information sheet containing the interview aims and participation details, including the right to withdraw from the research. Written informed consent was obtained from each participant. Interviews were conducted remotely (i.e. via phone, Zoom or Microsoft Teams). Interviews were audio-recorded

and reflective notes were taken during interviews. Interview participants received a £10 gift voucher as a thank you for their time and contribution.

Interviews followed a topic guide designed to explore participants' views, intentions and experiences of vaccinating against COVID-19, COVID-19 vaccine decision-making, sources of COVID-19 vaccine information, and recommendations on how to improve COVID-19 vaccination programme communication and delivery to H&SCWs.

Interviews were conducted by SB, PP, OOA, and SI between 10[th] February and 19[th] March 2021.

**Interview analysis.** Interviews were transcribed verbatim and analysed thematically by SB using the stages of data familiarization, coding, and theme identification and refinement [15]. To enhance the rigour of the analysis, coding approaches and subsequent theme generation were discussed by SB, PP, OOA, SI, RC, and SM-J. NVivo 12.0 software was used to manage the data and aid analysis.

# Findings

## Participants

Of the 1917 participants, 1656 (86.4%) were HCWs and 261 (13.6%) were SCWs. Most participants were female (n = 1446, 75.4%) and aged 35–64 years (n = 1494, 77.9%). The largest occupational groups among HCWs were registered nurses and midwives (n = 572; 34.5% of HCWs) and medical professionals (n = 430, 26.0% of HCWs). The highest proportions of SCWs were employed in domiciliary (i.e. home care) (n = 89, 34.1% of SCWs), community (n = 76, 29.1% of SCWs), or residential (n = 48, 18.4%) sectors.

1102 participants (57.5%) were White British or White Irish, 94 (4.9%) were from other White backgrounds, 168 (8.8%) were Black African or Mixed Black African, 66 (3.4%) were Black Caribbean or Mixed Black Caribbean, 264 (13.8%) were from an Indian background, 109 (5.7%) were from a South-East Asian (including Pakistani and Bangladeshi) or Mixed Asian background and 90 (4.7%) were not categorised into any of the six ethnicity categories. Each occupation varied by ethnicity (see Table 1).

1762 (91.9%) participants had been offered COVID-19 vaccination and, of those, 116 (6.6%) had declined the offer. See Tables 2 and 3 for a summary of COVID-19 vaccination offer and uptake by key demographic factors.

A third of survey participants (n = 640; 33.5%) provided their details to be contacted for interview. Most of these indicated in the survey that they had been vaccinated (n = 534; 83.4%) and were HCWs (n = 563; 88.0%). All participants that reported declining COVID-19 vaccination and left their contact details were contacted (n = 28). We also contacted 34 H&SCWs that reported they had been or planned to get vaccinated, 16 H&SCWs who had not yet been offered the vaccine, and 1 H&SCW who did not provide their vaccination status. In total, 81 participants were contacted, of these 16 HCWs and 4 SCWS were interviewed.

Participants were recruited from across ethnic groups, age categories, job roles, and geographical areas (see Table 4). In the period between the survey and the interview, three participants had changed their decision from declining to accepting COVID-19 vaccination. At interview all participants had been offered vaccination, 12 had been vaccinated, 1 participant had booked in to receive their vaccine, and 7 had declined vaccination.

## Quantitative findings: Associations with COVID-19 vaccination offer

107 participants (5.6%) had not been offered COVID-19 vaccination at the time of survey.

When controlling for other demographic and situational factors, Black African or Mixed Black African participants were approximately twice as likely to not be offered the vaccine

**Table 1. Frequency and percentage of sample within occupation group by ethnic minority grouping.**

| | White British and White Irish | White other | Black or Black British African or Mixed Black African | Black or Black British Caribbean or Mixed Black Caribbean | Asian or Asian British: Indian | South East Asian or Mixed Asian | Other minority ethnic groups | Prefer not to say |
|---|---|---|---|---|---|---|---|---|
| | N (% of occupation group) | N (% of occupation group) | N (% of occupation group) | N (% of occupation group) | N (% of occupation group) | N (% of occupation group) | N (% of occupation group) | N (% of occupation group) |
| **Healthcare** | | | | | | | | |
| Allied Health Professionals* | 152 (63.1%) | 16 (6.6%) | 17 (7.1%) | 8 (3.3%) | 23 (9.5%) | 10 (4.1%) | 11 (4.6%) | 4(1.7%) |
| Medical | 94 (21.9%) | 14 (3.3%) | 53 (12.3%) | 4 (0.9%) | 189 (44.0%) | 43 (10.0%) | 28 (6.5%) | 5 (1.2%) |
| Ambulance (operational) | 9 (90.0%) | 0 (0%) | 1 (10.0%) | 0 (0%) | 0 (0%) | 0 (0%) | 0 (0%) | 0 (0%) |
| Public Health | 19 (63.3%) | 1 (3.3%) | 3 (10.0%) | 2 (6.7%) | 1 (3.3%) | 2 (6.7%) | 2 (6.7%) | 0 (0%) |
| Commissioning | 4 (50.0%) | 0 (0%) | 1 (12.5%) | 2 (25.0%) | 0 (0%) | 1 (12.5%) | 0 (0%) | 0 (0%) |
| Registered Nurses and Midwives | 385 (67.3%) | 33 (5.8%) | 55 (9.6%) | 20 (3.5%) | 25 (4.4%) | 25 (4.4%) | 23 (4.0%) | 6 (1.0%) |
| Nursing or Healthcare Assistants | 135 (68.9%) | 12 (6.1%) | 10 (5.1%) | 9 (4.6%) | 9 (4.6%) | 11 (5.6%) | 9 (4.6%) | 1 (0.5%) |
| Wider Healthcare Team | 44 (62.0%) | 2 (2.8%) | 5 (7.0%) | 7 (9.9%) | 1 (1.4%) | 6 (8.5%) | 5 (7.0%) | 1 (1.4%) |
| General Management | 49 (52.1%) | 3 (3.2%) | 11 (11.7%) | 8 (8.5%) | 11 (11.7%) | 5 (5.3%) | 6 (6.4%) | 1 (1.1%) |
| Do not know/Did not answer | 4 (80.0%) | 0 (0%) | 0 (0%) | 1 (20.0%) | 0 (0%) | 0 (0%) | 0 (0%) | 0 (0%) |
| **Healthcare Total** | **895 (54.3%)** | **81 (4.9%)** | **156 (9.4%)** | **60 (3.6%)** | **259 (15.7%)** | **103 (6.2%)** | **84 (5.1%)** | **18 (1.1%)** |
| **Social care** | | | | | | | | |
| Residential | 36 (75.0%) | 1 (2.1%) | 2 (4.2%) | 2 (4.2%) | 2 (4.2%) | 2 (4.2%) | 1 (2.1%) | 2 (4.2%) |
| Domiciliary | 79 (88.8%) | 3 (3.4%) | 4 (4.5%) | 0 (0%) | 0 (0%) | 1 (1.1%) | 0 (0%) | 2 (2.2%) |
| Day | 6 (35.3%) | 3 (17.6%) | 2 (11.8%) | 2 (11.8%) | 0 (0%) | 0 (0%) | 3 (17.6%) | 1 (5.9%) |
| Community | 63 (82.9%) | 4 (5.3%) | 3 (3.9%) | 1 (1.3%) | 2 (2.6%) | 2 (2.6%) | 1 (1.3%) | 0 (0%) |
| Regulated Professionals | 6 (85.7%) | 0 (0%) | 0 (0%) | 0 (0%) | 0 (0%) | 0 (0%) | 1 (14.3%) | 0 (0%) |
| Other | 16 (69.6%) | 2 (8.7%) | 0 (0%) | 0 (0%) | 1 (4.3%) | 1 (4.3%) | 0 (0%) | 3 (13.0%) |
| Do not know/Did not answer | 1 (100%) | 0 (0%) | 0 (0%) | 0 (0%) | 0 (0%) | 0 (0%) | 0 (0%) | 0 (%) |
| **Social care Total** | **207 (79.1%)** | **13 (5.0%)** | **11 (4.2%)** | **5 (1.9%)** | **5 (1.9%)** | **6 (2.3%)** | **6 (2.3%)** | **8 (3.1%)** |

*Allied Health Professionals are healthcare professionals distinct from medicine and nursing. The roles of these professionals involve providing diagnostic, technical, therapeutic and support services in connection with healthcare.

than White British or White Irish participants (OR: 2.011, 95%CI: 1.026–3.943). Similarly, SCWs were also 50% more likely to not be offered vaccination than HCWs (OR: 1.453, 95%CI: 1.244–1.696).

In addition to these demographic findings, we found perception of the workplace environment to be associated with vaccine offer, with each extra point of agreement (out of 5) with the statement '*I would recommend my organisation as a place to work*' participants were approximately 30% (OR: 0.778; 95%CI: 0.640–0.947) more likely to be offered the vaccine. Participants that indicated a health issue or disability were also more likely to be offered the vaccine than those that did not indicate such issues (OR: 2.311; 95%CI: 1.232–4.334).

**Table 2. Participants demographic characteristics by being offered a vaccine and vaccine uptake.**

| | | Have you been offered a COVID-19 vaccine? | | | |
|---|---|---|---|---|---|
| | | No response | No | Yes | |
| | | | | Have you had, or are you intending to have the COVID-19 vaccine? | |
| Variable | N | | | Accepted or will accept | Declined or will decline |
| Full sample | 1917 | 48 (2.5%) | 107 (5.6%) | 1646 (85.9%) | 116 (6.1%) |
| **Gender** | | | | | |
| Male | 443 | 11 (2.5%) | 28 (6.3%) | 386 (87.1%) | 18 (4.1%) |
| Female | 1446 | 35 (2.4%) | 75 (5.2%) | 1240 (85.8%) | 96 (6.6%) |
| Prefer to self-describe | 12 | 0 (0%) | 1 (8.3) | 10 (83.3%) | 1 (8.3%) |
| No response | 16 | 2 (12.5%) | 3 (18.8) | 10 (62.5) | 1 (6.3%) |
| **Ethnicity** | | | | | |
| White British and White Irish | 1102 | 25 (2.3%) | 61 (5.5%) | 960 (87.1%) | 56 (5.1%) |
| White other | 94 | 4 (4.3%) | 6 (6.4%) | 76 (80.8%) | 8 (8.5%) |
| Black or Black British African or Mixed Black African | 168 | 6 (3.6%) | 15 (8.9%) | 131 (78.0%) | 16 (9.5%) |
| Black or Black British Caribbean or Mixed Black Caribbean | 66 | 1 (1.5%) | 2 (3.0%) | 48 (72.7%) | 15 (22.7%) |
| Asian or Asian British: Indian | 264 | 5 (1.9%) | 14 (5.3%) | 239 (90.5%) | 6 (2.3%) |
| Other South East Asian or Mixed Asian | 109 | 1 (0.9%) | 2 (1.8%) | 101 (92.7%) | 5 (4.6%) |
| Other ethnic minority groups | 90 | 2 (2.2%) | 2 (2.2%) | 81 (90.0%) | 5 (5.6%) |
| No response | 24 | 4 (16.7%) | 5 (20.8%) | 10 (41.6%) | 5 (20.8%) |
| **Age** | | | | | |
| Under 25 Years | 56 | 1 (1.8%) | 5 (8.9%) | 48 (85.7%) | 2 (3.6%) |
| 25 to 35 Years | 299 | 1 (0.3%) | 12 (4.0%) | 261 (87.3%) | 25 (8.4%) |
| 35 to 44 Years | 425 | 13 (3.1%) | 21 (4.9%) | 362 (85.2%) | 29 (6.8%) |
| 45 to 54 Years | 636 | 16 (2.5%) | 34 (5.3%) | 550 (86.5%) | 36 (5.7%) |
| 55 to 64 years | 433 | 12 (2.8%) | 30 (6.9%) | 370 (85.5%) | 21 (4.8%) |
| 65 years and over | 64 | 4 (6.3%) | 4 (6.3%) | 53 (82.8%) | 3 (4.7%) |
| No response | 4 | 1 (25.0%) | 1 (25.0%) | 2 (50.0%) | 0 (0%) |
| **Religion** | | | | | |
| Atheist/No Religion | 635 | 17 (2.7%) | 30 (4.7%) | 553 (87.1%) | 35 (5.5%) |
| Christian and all other Christian denominations | 900 | 22 (2.4%) | 56 (6.2%) | 762 (84.7%) | 60 (6.7%) |
| Hindu | 173 | 4 (2.3%) | 10 (5.8%) | 154 (89.0%) | 5 (2.9%) |
| Muslim | 93 | 1 (1.1%) | 3 (3.2%) | 85 (91.4%) | 4 (4.3%) |
| Other | 77 | 1 (1.3%) | 6 (7.8%) | 62 (80.5%) | 8 (10.4%) |
| No response | 39 | 3 (7.7%) | 2 (5.1%) | 30 (76.9%) | 4 (10.3) |
| **Long term health problem or disability** | | | | | |

*(Continued)*

**Table 2.** (Continued)

| | | Have you been offered a COVID-19 vaccine? | | | |
| | | No response | No | Yes | |
| | | | | Have you had, or are you intending to have the COVID-19 vaccine? | |
| Variable | N | | | Accepted or will accept | Declined or will decline |
|---|---|---|---|---|---|
| Yes, limited a little | 215 | 6 (2.8%) | 19 (8.8%) | 177 (82.3%) | 13 (6.0%) |
| Yes, limited a lot | 45 | 1 (2.2%) | 4 (8.9%) | 38 (84.4%) | 2 (4.4%) |
| No | 1625 | 38 (2.3) | 78 (4.8%) | 1411 (86.8%) | 98 (6.0%) |
| No response | 32 | 3 (9.4%) | 6 (18.8%) | 20 (62.5%) | 3 (9.4%) |

Finally, we also found an association between past seasonal influenza vaccination and being offered COVID-19 vaccination, with participants that reported receiving seasonal influenza vaccination in both 2019/20 and 2020/21 over twice as likely (OR: 0.371; 95%CI: 0.218–0.633) to have been offered COVID-19 vaccination than those that had not received the influenza vaccine in the last two years.

Table 5 gives coefficients and the Wald statistic, odds ratio and associated degrees of freedom for each of the predictor variables for COVID-19 vaccination offer.

## Quantitative findings: Associations with declining COVID-19 vaccination

116 participants (6.6%) declined COVID-19 vaccination when offered. Among the demographic variables, only ethnicity was a predictor of declining COVID-19 vaccination. After controlling for other variables in a multivariate model, Black African or Mixed Black African participants (OR: 5.550, 95%CI: 2.294–13.428) were significantly more likely to decline vaccination than those of White British or White Irish ethnicity. All other comparisons across ethnicity were non-significant within the model, however, Black Caribbean or Mixed Black Caribbean was significant at the univariate level.

Greater agreement with the statements '*I feel/felt under pressure from my employer to get a COVID-19 vaccine*' and '*I was/am worried about getting side-effects from a COVID-19 vaccine*' were associated with declining vaccination (OR: 1.751, 95%CI: 1.271–2.413 and OR: 1.68, 95% CI: 1.152–2.45 respectively), while indications that participants felt that COVID-19 vaccination was important, safe and effective were associated with participants being more likely to accept vaccination (OR: 0.515, 95%CI: 0.329–0.805). Previous influenza vaccine uptake over the last two years appears predictive of vaccine uptake within the model (OR: 0.227, 95%CI: 0.108–0.478).

Table 6 gives coefficients and the Wald statistic, odds ratio and associated degrees of freedom for each of the predictor variables for declining or intending to decline COVID-19 vaccination.

## Quantitative findings: Differences in beliefs by ethnicity

Table 7 indicates the differences in beliefs and trust in sources of information across the ethnic minority groups as compared to White British or White Irish participants.

Black Caribbean and Mixed Black Caribbean participants were significantly less likely to perceive COVID-19 vaccines as safe (p = .004), as important for H&SCWs to get to protect themselves (p = .001) and their families (p = .001), and as important to get back to normal (p < .001). Black Caribbean and Black African participants were significantly less likely to agree

**Table 3. Participants location and occupation by being offered a vaccine and vaccine uptake.**

| | | | | | Have you been offered a COVID-19 vaccine? | | |
|---|---|---|---|---|---|---|---|
| | | | | No response | No | Yes | |
| | | | | | | Have you had, or are you intending to have the COVID-19 vaccine? | |
| Variable | | | N | | | Accepted or will accept | Declined or will decline |
| Full sample | | | 1917 | 48 (2.5%) | 107 (5.6%) | 1646 (85.9%) | 116 (6.1%) |
| **Location** | | | | | | | |
| **England** | | | 1790 | 40 (2.2%) | 99 (5.5%) | 1548 (86.5%) | 103 (5.8%) |
| | | South East | 358 | 9 (2.5%) | 13 (3.6%) | 311 (86.9%) | 25 (7.0%) |
| | | Greater London | 414 | 8 (1.9%) | 21 (5.2%) | 363 (87.7%) | 22 (5.3%) |
| | | North West | 231 | 10 (4.3%) | 8 (3.5%) | 200 (86.6%) | 13 (5.6%) |
| | | East of England | 88 | 3 (3.4%) | 8 (9.1%) | 73 (83.0%) | 4 (4.5%) |
| | | West Midlands | 196 | 3 (1.5%) | 12 (6.1%) | 167 (85.2%) | 14 (7.1%) |
| | | South West | 106 | 1 (0.9%) | 7 (6.6%) | 87 (82.1%) | 11 (10.4%) |
| | | Yorkshire and the Humber | 127 | 5 (3.9%) | 12 (9.4%) | 105 (82.7%) | 5 (3.9%) |
| | | East Midlands | 191 | 2 (1.0%) | 11 (5.8%) | 173 (90.6%) | 5 (2.6%) |
| | | North East | 79 | 4 (5.1%) | 6 (7.6%) | 66 (83.5%) | 3 (3.8%) |
| **Scotland** | | | 59 | 0 (0%) | 6 (10.2%) | 45 (76.3%) | 8 (13.6%) |
| **Wales** | | | 45 | 1 (2.2%) | 1 (2.2%) | 39 (86.7%) | 4 (8.9%) |
| **Northern Ireland** | | | 15 | 1 (6.7%) | 0 (0%) | 13 (86.7%) | 1 (6.7%) |
| No response | | | 8 | 6 (75.0%) | 1 (12.5%) | 1 (12.5%) | 0 (0%) |
| **Healthcare Sector** | | | | | | | |
| | Allied Health Professionals | | 241 | 7 (2.9%) | 12 (5.0%) | 205 (85.1%) | 17 (7.1%) |
| | Medical | | 430 | 10 (2.3%) | 12 (2.8%) | 395 (91.9%) | 13 (3.0%) |
| | Ambulance (operational) | | 10 | 0 (0%) | 0 (0%) | 10 (100%) | 0 (0%) |
| | Public Health | | 30 | 2 (6.7%) | 7 (23.3%) | 20 (66.7%) | 1 (3.3%) |
| | Commissioning | | 8 | 0 (0%) | 2 (25.0%) | 5 (62.5%) | 1 (12.5%) |
| | Registered Nurses and Midwives | | 572 | 11 (1.9%) | 17 (3.0%) | 502 (87.8%) | 42 (7.3%) |
| | Nursing or Healthcare Assistants | | 196 | 6 (3.1%) | 17 (8.7%) | 160 (81.6%) | 13 (6.6%) |
| | Wider Healthcare Team | | 71 | 1 (1.4%) | 5 (7.0%) | 63 (88.7%) | 2 (2.8%) |
| | General Management | | 94 | 8 (8.5%) | 3 (3.2%) | 79 (84.0%) | 4 (4.3%) |
| | Other | | 4 | 1 (25.0%) | 0 (0%) | 3 (75.0%) | 0 (0%) |
| | **Healthcare Sector Total** | | 1656 | 46 (2.8%) | 75 (4.5%) | 1442 (87.1%) | 93 (5.6%) |
| **Social Care Sector** | | | | | | | |
| | Residential | | 48 | 1 (2.1%) | 4 (8.3%) | 36 (75.0%) | 7 (14.6%) |
| | Domiciliary | | 89 | 0 (0%) | 7 (7.9%) | 78 (87.6%) | 4 (4.5%) |
| | Day | | 17 | 0 (0%) | 3 (17.6%) | 12 (70.6%) | 2 (11.8%) |
| | Community | | 76 | 1 (1.3%) | 9 (11.8%) | 58 (76.3%) | 8 (10.5%) |
| | Regulated Professionals | | 7 | 0 (0%) | 1 (14.3%) | 5 (71.4%) | 1 (14.3%) |
| | Other | | 23 | 0 (0%) | 8 (34.8%) | 14 (60.9%) | 1 (4.3%) |
| | No response | | 1 | 0 (0%) | 0 (0%) | 1 (100%) | 0 (0%) |
| | **Social Care Sector Total** | | 261 | 2 (0.8%) | 32 (12.3%) | 204 (78.2%) | 23 (8.8%) |

**Table 4. Interview participant characteristics.**

| ID | Gender | Age (years) | Ethnicity | Religion | Job sector | Time spent in direct contact with service users | Covid-19 vaccination status |
|---|---|---|---|---|---|---|---|
| 1 | Male | 45–54 | Asian or Asian British: Indian | Hindu | Healthcare | Almost all | Vaccinated |
| 2 | Female | 45–54 | White: British | No religion | Social care | Almost all | Unvaccinated |
| 3 | Male | 45–54 | White: British | Christian | Social care | Almost all | Vaccinated |
| 4 | Female | 25–34 | Mixed: Other mixed background | Christian | Healthcare | Around half | Vaccinated |
| 5 | Female | 35–44 | White: British | Atheist | Healthcare | None | Unvaccinated |
| 6 | Female | 55–64 | Asian or Asian British: Other Asian background | Christian | Healthcare | None | Unvaccinated but booked in to get vaccination (Had declined at time of survey) |
| 7 | Female | 45–54 | White: British | No religion | Healthcare | Around half | Vaccinated (Had declined at time of survey) |
| 8 | Female | 25–34 | White: Other White background | Christian | Social care | Almost all | Unvaccinated |
| 9 | Male | 35–44 | White: Other White background | Christian | Social care | Less than half | Unvaccinated |
| 10 | Female | 35–44 | Asian or Asian British: Indian | Hindu | Healthcare | Less than half | Vaccinated |
| 11 | Female | 45–54 | White: British | No religion | Social care | Less than half | Unvaccinated |
| 12 | Female | 25–34 | White: British | No religion | Healthcare | Around half | Unvaccinated |
| 13 | Female | 45–54 | Black or Black British: African | Christian | Healthcare | Less than half | Vaccinated |
| 14 | Female | 45–54 | Black or Black British: Caribbean | Christian | Healthcare | Less than half | Vaccinated |
| 15 | Female | 25–34 | White: British | No religion | Healthcare | None | Unvaccinated |
| 16 | Female | 55–64 | White: British | Christian | Healthcare | Around half | Vaccinated |
| 17 | Female | 35–44 | Mixed: White and Black Caribbean | Christian | Healthcare | Around half | Vaccinated |
| 18 | Female | 55–64 | Asian or Asian British: Indian | No religion | Healthcare | Almost all | Vaccinated |
| 19 | Female | 35–44 | Black or Black British: African | Muslim | Healthcare | None | Vaccinated |
| 20 | Male | 55–64 | Asian or Asian British: Indian | Hindu | Healthcare | Around half | Vaccinated* (Had declined at time of survey) |

that their family/friends expect them to accept a vaccine ($p < .001$, $p < .001$ respectively) and were more likely to worry about side-effects ($p < .001$, $p = .036$ respectively).

Asian H&SCWs tended to hold more beliefs predictive of vaccine uptake (i.e. believing the vaccine is safe, effective, important etc.) than other ethnicities.

Agreement that health system information sources could be trusted for advice on COVID-19 vaccination was high among all participants surveyed (NHS = 87.4%, Public Health England = 90.0%, Health Professional = 80.7, and scientists involved in vaccine development = 87.5%. Selecting either agree or strongly agree to the statement). Although, Black Caribbean or Mixed Black Caribbean indicated less agreement that scientists ($p < .001$) and the NHS ($p = .002$) were trustworthy sources of information as compared to White British and White Irish participants.

**Table 5. A logistic regression analysis of not being offered a COVID-19 vaccination.**

| | | Univariable analysis. Original data | | | Multivariable analysis (all listed variables) Imputed continuous variables | | |
|---|---|---|---|---|---|---|---|
| **Variable** | | | | | | | |
| **Basic demographics** | | | | | Included in analysis: N = 1583 (Forward Likelihood Ratio) | | |
| | | Sig (*p*) | OR | 95% CI | Sig (*p*) | OR | 95% CI |
| **Ethnicity** | | | | | | | |
| | White British or White Irish ŧ | - | - | - | - | - | - |
| | White other | 0.690 | 1.193 | 0.501–2.841 | 0.92 | 1.057 | 0.358–3.119 |
| | Black or Black British African or Mixed Black African | 0.077 | 1.705 | 0.944–3.077 | 0.042 | 2.011 | 1.026–3.943 |
| | Black or Black British Caribbean or Mixed Black Caribbean | 0.385 | 0.530 | 0.127–2.219 | 0.09 | 0.174 | 0.023–1.317 |
| | Asian or Asian British: Indian | 0.879 | 0.955 | 0.525–1.735 | 0.339 | 1.432 | 0.686–2.988 |
| | Other South East Asian or Mixed Asian | 0.112 | 0.315 | 0.076–1.307 | 0.268 | 0.440 | 0.103–1.881 |
| | Other Minority | 0.194 | 0.388 | 0.093–1.616 | 0.602 | 0.678 | 0.158–2.918 |
| **Sector** | | | | | | | |
| | Healthcare | - | - | - | - | - | - |
| | Social Care | | | | < .001 | 1.453 | 1.244–1.696 |
| **I would recommend my organisation as a place of work** | | < .0001 | 0.72 | 0.61–0.851 | 0.012 | 0.778 | 0.640–.947 |
| **Are your day-to-day activities limited because of health problems or disabilities?** | | | | | | | |
| | Yes, limited a little | 0.014 | 1.927 | 1.142–3.253 | 0.009 | 2.311 | 1.232–4.334 |
| | Yes, limited a lot | 0.218 | 1.937 | .676–5.551 | 0.338 | 1.774 | 0.549–5.731 |
| | No ŧ | - | - | - | - | - | - |
| **Flu vaccine uptake** | | | | | | | |
| | No flu vaccine uptake ŧ | - | - | - | - | - | - |
| | One out of two flu vaccines taken | 0.068 | 0.6 | 0.346–1.038 | 0.169 | 0.641 | 0.340–1.208 |
| | Flu vaccine taken both years | < .0001 | 0.323 | 0.208–0.502 | < .001 | 0.371 | 0.218–0.633 |

ŧ Comparison group.

## Survey free-text findings: Reasons for accepting COVID-19 vaccination

Of the 1731 participants that reported that they had either received or would likely accept COVID-19 vaccination if offered, 1577 (91%) provided a reason(s) for vaccine acceptance.

Fifty-six percent of participants wanted to be vaccinated to protect themselves/avoid catching COVID-19 (see Fig 1). The next most given reasons to accept vaccination were to protect family and/or friends (40% of participants), to protect service users (28%), to protect others/society/the community (24%), to get life back to 'normal', to control the pandemic (20%), and to protect colleagues (10%).

## Survey free-text findings: Reasons for declining COVID-19 vaccination

Of the 137 participants that had either declined or would likely decline COVID-19 vaccination if offered, 130 (95%) provided a reason for this. Fifty-one percent of participants were concerned about vaccine side effects and half of the participants expressed concerns around a lack of research. Twenty-seven percent of participants were concerned about the effectiveness of the vaccine, 21% did not feel at risk of severe COVID, and 21% distrusted government, the pharmaceutical industry, vaccine manufacturers and/or the media (see Fig 2). Fourteen percent were concerned about the vaccine development process being rushed and 11% were

**Table 6. A logistic regression analysis of declining or intending to decline COVID-19 vaccination.**

| | | Univariable analysis. Original data | | | Multivariable analysis (all variables) Imputed continuous variables | | |
|---|---|---|---|---|---|---|---|
| **Variable** | | | | | | | |
| **Basic demographics** | | | | | Included in analysis: N = 1495 (Forward Likelihood Ratio) | | |
| | | Sig (*p*) | OR | 95% CI | Sig (*p*) | OR | 95% CI |
| **Ethnicity** | | | | | | | |
| | White British and White Irish ŧ | | | | | | |
| | White other | 0.0136 | 1.805 | 0.830–3.924 | 0.984 | 1.015 | 0.243–4.235 |
| | Black or Black British African or Mixed Black African | 0.013 | 2.094 | 1.167–3.758 | < .001 | 5.550 | 2.294–13.428 |
| | Black or Black British Caribbean or Mixed Black Caribbean | < .001 | 5.357 | 2.827–10.153 | 0.260 | 1.880 | 0.627–5.637 |
| | Asian or Asian British:- Indian | 0.053 | 0.43 | 0.183–1.011 | 0.202 | 2.449 | 0.619–9.695 |
| | Other South East Asian or Mixed Asian | 0.732 | 0.849 | 0.332–2.167 | 0.921 | 0.929 | 0.217–3.98 |
| | Other Minority | 0.906 | 1.058 | 0.412–2.716 | 0.344 | 1.940 | 0.492–7.659 |
| **Flu vaccine uptake** | | | | | | | |
| | No flu vaccine uptake ŧ | | | | | | |
| | One out of two flu vaccines taken | < .001 | 0.226 | 1.32–.388 | 0.093 | 0.493 | 0.216–1.124 |
| | Flu vaccine taken both years | < .001 | 0.06 | 0.036–0.1 | < .001 | 0.227 | 0.108–0.478 |
| Combined COVID-19 vaccine beliefs (important, safe, and effective) | | < .001 | 0.054 | 0.036–0.083 | 0.004 | 0.515 | 0.329–0.805 |
| I felt under pressure from my employer to get a COVID-19 vaccine | | < .001 | 2.872 | 2.347–3.515 | 0.001 | 1.751 | 1.271–2.413 |
| I am worried about getting side-effects from a COVID-19 vaccine | | < .001 | 0.518 | 0.416–0.644 | 0.007 | 1.68 | 1.152–2.45 |

ŧ Comparison group.

concerned about the change in the maximum interval between first and second vaccine doses from 3 to 12 weeks. Eight percent reported having had COVID-19 as one of their main reasons for declining vaccination.

By ethnicity, Black African or Mixed Black African participants were more likely to state concerns about a lack of research (67%) and Indian participants were more likely to state concerns about side effects (71%). White-other participants were more likely to state concerns about effectiveness (55%) and White-British and Irish participants were more likely to not feel at risk of severe COVID-19 (36%). Black Caribbean or Mixed Black Caribbean participants were more likely to state distrust in government, the pharmaceutical industry, vaccine manufacturers and/or the media (38%).

SCWs were more likely to state concerns about effectiveness (37%), not feeling at risk of severe COVID-19 (37%), and a preference to build the immune system naturally (19%), as reasons for not vaccinating, compared with HCWs. None of the SCWs that completed the open text reported being concerned about the change in dosing schedule as their main reason for not vaccinating.

## Qualitative findings–from interviews and free-text responses

Factors influencing COVID-19 vaccine uptake amongst H&SCWs are presented under ten themes: (i) access; (ii) perceptions of COVID-19 risk and severity; (iii) trust; (iv) beliefs around vaccine effectiveness; (v) perceptions of COVID-19 vaccine importance; (vi) pressure to get vaccinated; (vii) concerns about allergies; (viii) vaccination concern in women of childbearing age; (ix) vaccine communication and information sources; and (x) religion.

**Table 7. COVID-19 belief and trust in information sources compared across ethnicity.**

| Statement | White British and White Irish | White other | | Black or Black British African or Mixed Black African | | Black or Black British Caribbean or Mixed Black Caribbean | | Asian or Asian British: Indian | | South East Asian or Mixed Asian | | Other Ethnic Groups | |
|---|---|---|---|---|---|---|---|---|---|---|---|---|---|
| Please indicate your level of agreement with each of the following statements: | Mean (SD) | Mean (SD) | (p) | Mean (SD) | (p) | Mean (SD) | (p) | Mean (SD) | (p) | Mean (SD) | (p) | Mean (SD) | (p) |
| "I think COVID-19 is deadlier than seasonal flu" | 3.70 (.589) | 3.59 (.789) | .7 | 3.77 (.545) | .784 | 3.68 (.636) | 1.0 | 3.82 (.480) | .035 | 3.84 (.455) | .189 | 3.81 (.423) | .662 |
| "I think it's important for social/health care workers to get a COVID-19 vaccine to protect themselves" | 3.79 (.583) | 3.67 (.777) | .497 | 3.78 (.512) | 1.0 | 3.47 (.826) | .001 | 3.93 (.304) | .015 | 3.83 (.468) | .999 | 3.82 (.445) | 1.0 |
| "I think it's important for social/health care workers to get a COVID-19 vaccine to protect their families" | 3.76 (.630) | 3.65 (.803) | .692 | 3.80 (.476) | .998 | 3.43 (.871) | .001 | 3.92 (.315) | .004 | 3.81 (.516) | .992 | 3.79 (.461) | 1.0 |
| "I think it's important for social/health care workers to get a COVID-19 vaccine to protect their patients" | 3.79 (.595) | 3.73 (.714) | .970 | 3.81 (.468) | 1.0 | 3.61 (.652) | .212 | 3.92 (.285) | .021 | 3.82 (.508) | .999 | 3.83 (.411) | 1.0 |
| "I think that COVID-19 vaccines are safe" | 3.48 (.689) | 3.33 (.839) | .468 | 3.38 (.653) | .656 | 3.07 (.959) | .004 | 3.67 (.538) | .005 | 3.56 (.632) | .961 | 3.42 (.635) | .995 |
| "I think that COVID-19 vaccines are effective" | 3.32 (.691) | 3.18 (.816) | .743 | 3.20 (.662) | .523 | 3.08 (.818) | .390 | 3.48 (.548) | .015 | 3.41 (.629) | .874 | 3.35 (.537) | 1.0 |
| "I think it is important for people to get vaccinated against COVID-19 to get life back to 'normal'" | 3.64 (.688) | 3.42 (.864) | .099 | 3.52 (.649) | .462 | 3.21 (.967) | < .001 | 3.70 (.555) | .938 | 3.61 (.698) | 1.0 | 3.54 (.646) | .909 |
| "I think that COVID-19 vaccines are compatible with my religious beliefs"* | 3.74 (.640) | 3.52 (.839) | .301 | 3.47 (.817) | < .001 | 3.27 (,917) | < .001 | 3.87 (.435) | .19 | 3.74 (.594) | 1.0 | 3.58 (.766) | .01 |
| "I feel well informed about COVID-19 vaccination" | 3.47 (.745) | 3.19 (.965) | .017 | 3.36 (.778) | .678 | 2.86 (1.021) | < .001 | 3.64 (.614) | .023 | 3.41 (.772) | .997 | 3.35 (.785) | .847 |
| "My family and friends expect me to accept a COVID-19 vaccine" | 3.57 (.711) | 3.21 (.984) | .003 | 3.03 (1.020) | < .001 | 2.60 (1.213) | < .001 | 3.72 (.593) | .117 | 3.32 (.915) | .049 | 3.35 (.797) | .260 |
| "My colleagues expect me to accept a COVID-19 vaccine" | 3.52 (.682) | 3.33 (.763) | .236 | 3.45 (.765) | .937 | 3.39 (.802) | .865 | 3.66 (.646) | .118 | 3.61 (.632) | .921 | 3.52 (.685) | 1.0 |
| "I felt under pressure from my employer to get a COVID-19 vaccine" | 1.93 (.984) | 2.19 (1.063) | .279 | 2.03 (1.072) | .943 | 2.32 (1.238) | .061 | 1.61 (.879) | < .001 | 1.97 (1.057) | 1.0 | 1.99 (1.006) | 1.0 |
| "I am worried about getting side-effects from a COVID-19 vaccine" | 2.32 (1.003) | 2.52 (1.068) | .681 | 2.59 (1.076) | .036 | 2.92 (1.168) | < .001 | 2.07 (.988) | .006 | 2.54 (1.093) | .417 | 2.54 (1.067) | .555 |
| **I trust the advice on Covid-19 vaccination given by. . .** | | | | | | | | | | | | | |
| My work colleagues | 3.69 (1.005) | 3.31 (1.180) | .012 | 3.69 (1.028) | 1.0 | 3.36 (1.032) | .173 | 4.03 (.902) | < .001 | 3.82 (.964) | .911 | 3.69 (1.029) | 1.0 |
| Social media | 2.30 (.960) | 2.28 (1.072) | 1.0 | 2.28 (1.011) | 1.0 | 2.37 (1.025) | 1.0 | 2.64 (1.054) | < .001 | 2.49 (1.055) | .548 | 2.40 (1.147) | .989 |
| Community leaders | 3.11 (.952) | 2.84 (.969) | .170 | 3.03 (.959) | .978 | 2.90 (.933) | .702 | 2.90 (.974) | .033 | 3.01 (.931) | .965 | 3.15 (.963) | 1.0 |
| Religious leaders | 2.69 (.882) | 2.28 (.958) | .003 | 2.92 (1.084) | .110 | 2.88 (.892) | .808 | 2.67 (1.036) | 1.0 | 2.84 (1.102) | .801 | 2.83 (1.004) | .911 |
| NHS | 4.38 (.879) | 4.17 (.920) | .368 | 4.23 (.836) | .470 | 3.92 (1.085) | .002 | 4.30 (.854) | .916 | 4.35 (.825) | 1.0 | 4.23 (.762) | .815 |
| News media (e.g. print or online newspapers, radio, and television news broadcasts) | 2.92 (1.030) | 2.87 (1.129) | 1.0 | 3.25 (1.002) | .006 | 2.88 (1.075) | 1.0 | 3.30 (.948) | < .001 | 2.84 (1.097) | .994 | 3.12 (.986) | .681 |
| Government | 3.40 (1.163) | 2.95 (1.242) | .010 | 3.50 (1.166) | .960 | 2.80 (1.166) | .002 | 3.43 (1.150) | 1.0 | 3.14 (1.251) | .344 | 3.36 (1.131) | 1.0 |
| Family | 3.34 (.915) | 3.26 (.983) | .994 | 3.45 (1.037) | .873 | 3.21 (1.018) | .972 | 3.69 (.908) | < .001 | 3.20 (1.074) | .850 | 3.20 (.907) | .917 |

*(Continued)*

**Table 7.** (Continued)

| Statement | White British and White Irish | White other | | Black or Black British African or Mixed Black African | | Black or Black British Caribbean or Mixed Black Caribbean | | Asian or Asian British: Indian | | South East Asian or Mixed Asian | | Other Ethnic Groups | |
|---|---|---|---|---|---|---|---|---|---|---|---|---|---|
| Please indicate your level of agreement with each of the following statements: | Mean (SD) | Mean (SD) | (*p*) | Mean (SD) | (*p*) | Mean (SD) | (*p*) | Mean (SD) | (*p*) | Mean (SD) | (*p*) | Mean (SD) | (*p*) |
| *Friends* | 3.26 (.869) | 3.24 (.964) | 1.0 | 3.38 (1.015) | .782 | 3.15 (.980) | .978 | 3.67 (.829) | < .001 | 3.30 (.987) | 1.0 | 3.25 (.879) | 1.0 |
| *Scientists involved in COVID-19 vaccine development* | 4.48 (.803) | 4.52 (.913) | 1.0 | 4.28 (.896) | .108 | 3.95 (1.175) | < .001 | 4.50 (.762) | 1.0 | 4.43 (.731) | .999 | 4.37 (.822) | .952 |
| *Public Health England* | 4.21 (.995) | 3.98 (1.164) | .416 | 4.12 (.910) | .962 | 3.82 (1.124) | .062 | 4.00 (1.080) | .065 | 4.23 (.959) | 1.0 | 4.21 (.865) | 1.0 |
| *Health Professionals* | 4.31 (.802) | 4.02 (1.114) | .030 | 4.22 (.801) | .881 | 3.97 (.894) | .028 | 4.41 (.699) | .576 | 4.38 (.624) | .982 | 4.25 (.713) | .999 |

* This statement was only presented to those participants that stated an option other than atheism or no religion (N = 1225).

We highlight where specific themes were reported more dominantly by HCWs or SCWs. We use the term participants when themes emerged in interviews and survey free-text responses, and highlight where themes emerged in interviews or free-text responses only.

**Access.** Most NHS HCWs reported that it was easy to organise vaccination through formal invitations via their employer. For others, including SCWs and those with non-mainstream NHS employers (private, bank staff, locums, pharmacists) accessing COVID-19 vaccination was described as 'quite a battle' as they were not given advice about how to organise vaccination by their employer(s), or received mixed messages around whether they should arrange an appointment through their general practitioner or employer. In trying to access

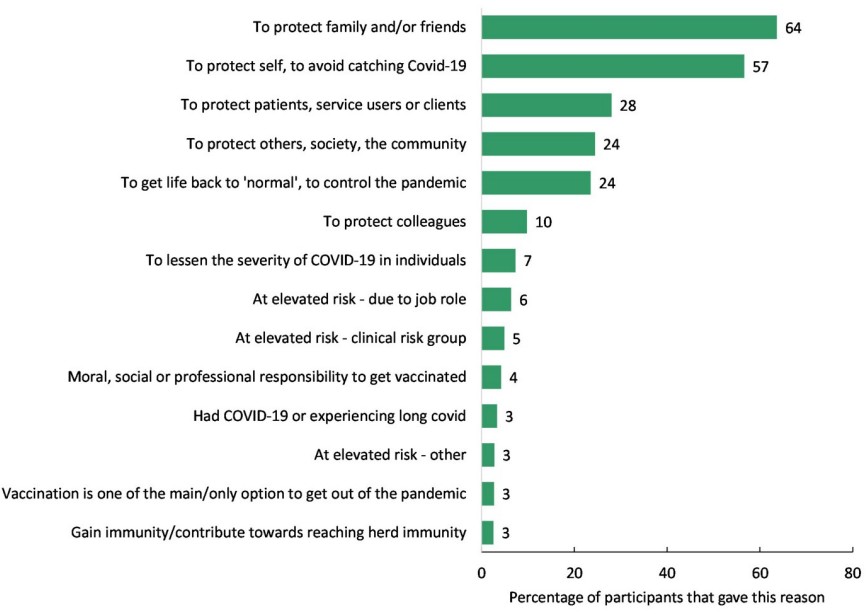

**Fig 1. Reasons for vaccinating.**

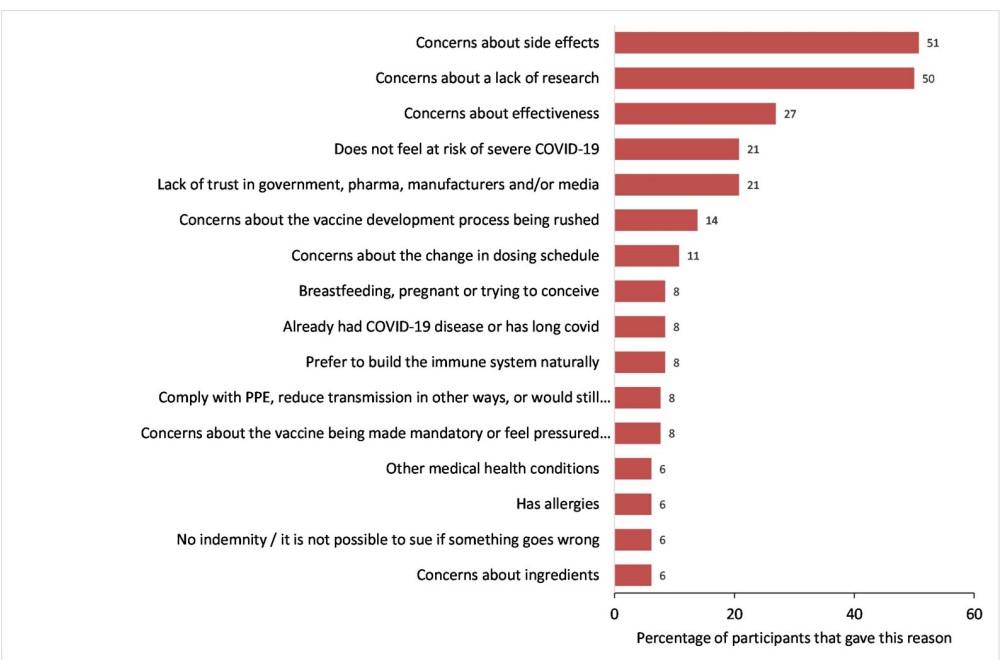

**Fig 2. Reasons for not vaccinating.**

COVID-19 vaccination, some of these HCWs had reached out to their general practice but were not able to access the vaccine through this route. One interviewee was told to wait until their age category joined the priority list to be vaccinated, and in one case was advised to book vaccination through an NHS web link.

One interviewed HCW felt that although some were able to access vaccination appointments during working hours, in practice this was not happening for all staff, noting that staff in job grades 5 and below, often belonging to ethnic minority groups, were less likely to have their time freed up during working hours to access vaccination.

**Perceptions of COVID-19 risk and severity.** How participants perceived their personal risk of COVID-19 was a major influence on vaccination decision-making. Most vaccinated participants considered themselves at risk of COVID-19 exposure, i.e. because of working in frontline roles, and/or at risk of severe COVID-19 due to one or a combination of factors including older age, clinical vulnerability, and being from an ethnic minority background. In interviews, participants who perceived themselves to be at higher risk of COVID-19 elaborated that this sense of personal risk outweighed vaccine safety concerns. Several survey participants felt that family members of H&SCWs should also have been offered vaccination as a priority.

Most unvaccinated participants considered themselves to be at low risk of COVID-19. These participants included younger adults and those without underlying health conditions, H&SCWs with limited/no service user contact at the time of vaccine offer, and those not living with household members at higher risk of COVID-19. One interviewee considered that gaining natural immunity through infection was better than vaccinating for those not in higher risk groups.

> 'So, for me, it's weighing up the pros and cons for different risks, doing my own risk assessment. I understand, you know, everybody has to do their own risk assessments, based on their own circumstances. And, for me, there are too many unknowns about this for somebody like

*me who is at very, very, little risk of serious illness.' (Interviewee #2 –SCW, female, White British)*

Vaccinated interviewees discussed COVID-19 as a severe disease and had often witnessed serious and life-threatening COVID-19 symptoms and complications first-hand (e.g. through being badly affected by COVID-19 themselves or seeing others with severe COVID-19).

Several participants that had declined vaccination considered COVID-19 severity as over-played by the government or the media (with an over-representation of cases in younger people), or considered the disease severe for others (e.g. people in clinically vulnerable groups) but not for themselves. These views were also based on some having had COVID-19 but only experiencing mild symptoms, or not knowing people that had been affected.

Two interviewees (one HCW and one SCW) expressed scepticism around how COVID-19 deaths are reported, questioning how many of these might be linked to non-COVID-19 causes exacerbated by disruptions to NHS care.

**Trust.**  Vaccination decisions were influenced by trust at multiple levels: in the vaccine, the provider, and policymakers. In particular, lower levels of trust at each level were reported amongst H&SCWs from Black and Asian ethnic minority groups where trust was undermined by beliefs in systemic racism (e.g. racism in healthcare, in medical research, in government). SCWs also reported lower levels of trust, which appeared more linked to relationships with their employer and feeling pressurised to vaccinate.

*Trust in the vaccine.* Vaccinated and unvaccinated participants expressed concern over potential COVID-19 vaccine side effects, particularly long-term effects. Many participants were concerned about the speed of vaccine development and delivery, stating that COVID-19 vaccines remain in the trial phase and there is a lack of long-term outcome data. Some participants reported that they or their colleagues had wanted to 'wait and see' how the vaccination worked in others before getting vaccinated themselves. In free-text responses, several HCWs noted that the indemnification of the Pfizer vaccine (i.e. protecting Pfizer from civil lawsuits in the event of unforeseen vaccine complications) also undermined trust in the vaccine.

Questions were raised about whether vaccines might interact with medications, how vaccination might affect people who already had COVID-19, and the safety of COVID-19 vaccines and of mRNA vaccines in particular. These views were expressed in interviews and free-text responses.

*'I have had long Covid and I cannot get enough information to support me getting this information that taking this vaccine will not exacerbate my symptoms and ill health, to feel safe about this choice. I feel totally lost in this situation. Information is bouncing around a 1,000-miles an hour and I feel pushed to make an uninformed decision and it is totally overwhelming' (Survey ID387– HCW, female, White Other)*

Concerns were raised that COVID-19 vaccines had not been sufficiently trialled in ethnic minority groups, and therefore there was insufficient safety and effectiveness data for use in people from ethnic minority backgrounds. One interviewee, of Indian ethnicity, spoke specifically about wanting a particular vaccine because it had been trialled in India, and had wanted to wait for this vaccine to become available before vaccinating.

*Trust in the provider.* Participants raised concerns about past medical racism (e.g. the Tuskegee syphilis study) as being commonly reported amongst colleagues from Black ethnic groups. Some pointed to reports of unethical medical experimentation by Pfizer in Africa in the past, and one interviewee reported that staff had been given the option of accessing the AstraZeneca vaccine, which was available at a different site.

Several participants reported that the medical racism they or others within their ethnic group had experienced in the past fuelled their vaccine hesitancy, such as Black women being treated adversely by healthcare providers. Two survey participants voiced scepticism in being asked to have data recorded around ethnicity prior to vaccination, as they were concerned this might influence the vaccine given to them.

Participants voiced their distrust in the pharmaceutical industry and scientists, drawing on examples of harms caused by medicines and vaccines introduced in the past, such as swine flu vaccination (linked to increased risk of narcolepsy [16, 17]), which also undermined trust in COVID-19 vaccines.

*Trust in policymakers*. Several interviewees and survey participants reported that distrust in the government and its handling of the pandemic had affected their own or colleagues' trust in vaccination. Amongst unvaccinated interview participants, notably SCWs, questions were raised around the length of time NHS services had been restricted and reduced, the changing guidance e.g. on mask wearing, and the motivation for continual lockdowns.

Participants shared their frustrations about changes to the interval between vaccine doses, against the manufacturers recommended protocol, and for some this negatively affected COVID-19 vaccination confidence and left them feeling vulnerable. Several participants felt the dosing interval change was not well communicated. Other aspects around COVID-19 vaccine delivery undermined confidence e.g. talks about mixing COVID-19 vaccines, and differences in vaccination programme delivery between countries.

Participants reported how the pandemic had hit ethnic minority groups hardest, and how poorly vaccination and ethnicity risk had been communicated. In free-text responses, a number of Black ethnicity participants voiced scepticism around the sudden drive to protect ethnic minorities from COVID-19, given the 'decades long' health inequalities experienced by ethnic minority groups. Several participants considered racism to be deeply embedded within government, leading them to question why people from ethnic minority backgrounds would trust government recommendations on vaccination.

> *'We feel that for decades, we have been on the receiving end of health and social inequalities. Now, we are being presented with a new vaccine that has been developed in quick time and we are now priority group number 1' (Survey ID52 –HCW, male, Black or Black British:— African)*

A small number of participants voiced some critique around the use of the collective term BAME (Black, Asian, and Minority Ethnic), which does not highlight heterogeneity between or within groups and may mask particular inequalities. One participant reported: '*I'm tired of people talking about those BAME people without understanding that they're different groups, and without trying to understand why.*' (Interviewee#14 –HCW, female, Black or Black British: —Caribbean).

**Beliefs around vaccine effectiveness.** Most vaccinated participants felt confident that vaccination would reduce their risk of contracting COVID-19 and, if they became infected, their likelihood of developing severe disease and/or complications. One commonly reported factor influencing decision-making was the uncertainty around COVID-19 vaccine effectiveness in reducing transmission. Without information on this, where people felt at low personal risk, there was little impetus to get vaccinated.

> *'I always said if the vaccine doesn't stop transmission, what is the point of me having it? That was one of my early sort of arguments against it, was I'm not injecting something into me if*

*it's not going to do the job that they say it's going to do.' (Interviewee #7 –HCW, female, White British)*

Several survey participants were concerned that the effectiveness of COVID-19 vaccination may be undermined by new variants of COVID-19.

**Perceptions of COVID-19 vaccine importance.** Vaccinated participants reported that they or their colleagues viewed vaccination as important to protect themselves and others, including clinically vulnerable family members, and also as a route out of the pandemic and a return to 'normality' (e.g. being able to travel and visit loved ones). Several vaccinated participants considered vaccination necessary for them to keep being able to work. One participant also considered vaccination important to protect against newer strains of COVID-19.

Unvaccinated participants tended to consider vaccination as less important. One interview participant and several survey participants felt being vaccinated would not impact on their need to wear personal protective equipment or make a difference in terms of their ability to see loved ones.

**Pressure to get vaccinated.** Several participants, mainly SCWs, felt vaccination was not presented as a choice. Pressure was mostly discussed as coming from employers; however, one participant, living with older relatives, also felt pressured from family.

Amongst unvaccinated participants, concerns were raised about how their decision might impact their freedoms (i.e. ability to access certain public spaces or travel) and most worryingly their job security. For SCWs, pressure was exacerbated by hearing of care sector employers making COVID-19 vaccination mandatory for staff, and the vulnerability of SCW positions (e.g. employment on zero-hours contracts).

*'I am disgusted by the narrative that it is some sort of civic duty to take the vaccine. It should always be the free choice of the individual based upon their own health situation. I have deep concerns at the potential for coercion regarding vaccination which we are already witnessing. I know care home staff who have been threatened with dismissal should they not be vaccinated and have ended up having the vaccine despite not wanting to as they felt they would lose their jobs.' (Survey ID663 –SCW, female, White British)*

Participants were alarmed that their vaccination decisions were relayed to managers when health related issues would usually be supported by occupational health or general practitioners only. Several participants reported that managers had access to lists of unvaccinated staff and were contacting them directly to discuss their vaccination status.

Feeling pressurised had damaging effects, eroding trust and negatively affecting relationships at work. It cemented several participant's stances on declining vaccination, making them more vaccine hesitant, and pushed one person into having the vaccine when they would have preferred not to. Participants strongly felt that vaccination should be an individual's choice, informed and voluntary.

**Concerns about allergies.** Several participants voiced concerns about allergies and a lack of early information on their suitability for vaccination. One unvaccinated interviewee had concerns about having had a severe adverse reaction to a past vaccination while another had delayed vaccination until a specific vaccine became available, which they were advised was more suitable for people with allergies.

**Vaccination concern in women of childbearing age.** Two interviewees and several survey participants who wanted to become pregnant in the future were concerned about unfounded rumours that COVID-19 vaccines cause fertility problems. Another interviewee was concerned about the safety of COVID-19 vaccination and breastfeeding and had decided

to delay vaccination. Updated guidance on vaccinating during pregnancy enhanced uncertainty amongst some pregnant women about vaccinating.

**Vaccine communication and information sources.**   Access to information varied amongst staff, with differences reported across job roles and also by ethnicity, and this affected beliefs and behaviours. Participants recommended that information should be communicated regularly by employers across various platforms to reach different audiences e.g. email, social media, webinars. One recommendation, voiced by several interview and survey participants, was that engagement with faith groups was particularly important in communicating with staff from ethnic minority backgrounds. Participants also wanted more openness in communication about uncertainty, so they could make an informed decision around vaccination.

All interviewees reported actively searching for COVID-19 vaccine information and most discussed vaccination with colleagues. Participants tended to avoid social media sources in favour of accessing government or NHS sources. Many participants preferred to access research articles directly, with concerns that second-hand reporting of the information (i.e. in the media) may be incomplete or biased.

Unvaccinated participants reported that although they might receive information about vaccination, they were not given the opportunity to have any discussion around vaccine questions and concerns. Instead, they felt they were dismissed as being anti-vaccination and treated as stupid, something they felt was exacerbated by some of the terminology around vaccination, including anti-vax and myth busting. One participant voiced that Black people have historically been silenced and told not to ask questions around healthcare, and that this was happening with COVID-19 vaccination and undermining trust.

*'I wanted to be fully informed rather than just told to take it. . . I was left frightened and with little information which was totally avoidable.' (Survey ID350 –HCW, female, Mixed:— White and Black Caribbean)*

Participants said that it was difficult to keep up-to-date and understand reasons for changes to the COVID-19 vaccination programme e.g. updated guidance on vaccination in pregnancy. There were also reports of participants being given mixed messages about vaccination. For example, two interviewees that had declined vaccination had been given different information from either healthcare professionals or their employer on vaccines e.g. advised that younger people should access a certain vaccine. Finally, many survey participants also suggested that more transparency was needed about side effects, which they felt had been downplayed.

*'I would like to know more about side effects. Besides what is online there appears to be a vagueness about the side effects that I'm sure some people have had. I think there should be more transparency especially for Black and brown healthcare workers.' (Survey ID872 – HCW, female, Black or Black British:—Other)*

**Religion.**   For one interviewee, it was essential that vaccination was halal–permissible in Islam–and they sought reassuring information about COVID-19 vaccination through Islamic medical groups and journals. Views were also expressed in free-text responses about vaccinating during Ramadan. Muslim faith leaders have advised that COVID-19 vaccination is permissible during Ramadan and does not nullify the fast.

*'I have also decided to postpone having my first vaccine because my second vaccine would fall in Ramadan and I will not miss my fasting'. (Survey ID234—HCW, female, Asian or Asian British:—Pakistani)*

## Discussion

COVID-19 vaccination uptake was high amongst our participants (93.9% of HCWs and 90.0% of SCWs were offered and accepted vaccination); however, we identified variations in vaccination offer and uptake by job sector and ethnicity.

We found that SCWs were offered COVID-19 vaccination at a lower rate than HCWs (87.0% *vs* 92.7%) and that SCWs were less clear about who was responsible for their COVID-19 vaccination offer; whether GP or employer. This is likely to be linked to the organisational structure and nature of roles within social care. Social care can be provided by local authorities, private sector companies or voluntary organisations. Funding is either paid for by the individual or by local authorities where the individual cannot self -fund. Care is often provided in small care home settings or in an individual's own home so the workforce is more disperse. In addition, workforce vaccination as part of the seasonal influenza vaccination programme is more embedded within the NHS than social care and therefore NHS organisations may be better prepared for COVID-19 vaccination delivery. The effectiveness of the SCW influenza vaccination programme is arguably also undermined by conflicting guidance on who should provide the vaccination. Since 2017/18 the SCW workforce has been able to access seasonal influenza vaccination freely through primary care but this service is 'intended to complement, not replace, any established occupational health schemes that employers have in place' [18]. Influenza vaccination is not consistently monitored amongst SCWs and uptake remains low [19]. In our study, in several instances, the onus appeared to be on SCWs to organise vaccination, rather than on their employers. This was also reported by interviewed non-mainstream NHS HCWs, suggesting that outside of the NHS greater barriers to vaccine offer and access may be experienced.

Our analysis indicated that Black African or Mixed Black African participants were being offered COVID-19 vaccination at a lower rate than White British and White Irish participants (87.5% *vs* 92.1%). This appeared as a significant finding in our multivariate regression analysis (OR:2.59, 95%CI: 1.292–5.177) and is difficult to explain and requires further investigation. Compared to White British and White Irish participants, we also found that rates of COVID-19 vaccination decline were higher amongst Black African or Mixed Black African participants and Black Caribbean or Mixed Black Caribbean participants. These findings are similar to data from an NHS Trust in England, which reported differential uptake by ethnicity (70.9% in White staff v 58.5% in South Asian and 36.8% in Black staff; $p < .001$) [7], and are also supported by international data showing lower intention to take up COVID-19 vaccination among healthcare workers from certain ethnic groups [20]. Comparison by ethnicity is further demonstrated by the interaction between ethnicity and COVID-19 vaccine beliefs, showing for example, less confidence in vaccine safety and importance amongst Black H&SCWs.

Through free-text responses and interviews we found that factors influencing vaccination-decisions were multi-layered and often involved the weighing up of perceived risks and benefits of vaccination. Factors found to influence COVID-19 vaccine uptake amongst H&SCWs included perceptions around COVID-19 risk and COVID-19 severity, beliefs around vaccine effectiveness, safety and importance, concerns about allergies, concerns about safety in women of childbearing age, and religious beliefs. Interviewees discussed that perceptions around personal COVID-19 risk and the severity of COVID-19 were central to decision-making, and

younger participants and/or those without underlying health conditions did not necessarily see the rationale for vaccinating, particularly without evidence on the link between vaccination and reducing disease transmission. Concerns around safety–especially in light of the speed of development of the new vaccines–also accord with findings from attitudinal work conducted internationally [21].

Distrust, reported in the vaccine, the provider and the policymaker, was also central to vaccination decisions. Distrust was particularly expressed by ethnic minority groups, who emphasised how they have borne the brunt of health inequalities before and during the pandemic. Black participants in particular felt stigmatised and patronised, and unable to voice questions and concerns, and obtain responses. This is consistent with existing evidence [22] and a recent PHE report [23]. Participants criticised the media characterisation of ethnic minority groups as anti-vax, which fails to recognise underlying reasons for lower vaccine confidence.

Organisational factors and workplace culture played an important role in the likelihood of both being offered and getting vaccinated, as described elsewhere [24]. This is consistent with previous research that indicates that an organisational culture framing seasonal influenza vaccination within a broader staff wellbeing programme was conducive to higher uptake [24, 25]. The coherence between staff well-being and COVID-19 vaccination might be particularly important in a context where staff have been subjected to high levels of stress over a long period [26]. Importantly, the survey revealed that participants that reported greater agreement with the statement *'I feel/felt under pressure from my employer to get a COVID-19 vaccine'* were significantly more likely to decline the vaccine even after demographic factors were controlled for. Interviews suggest placing staff under pressure to vaccinate may increase intention to refuse the vaccine. This was particularly evident in SCWs, for whom pressure was exacerbated by hearing of care sector employers making COVID-19 vaccination mandatory for staff, and the vulnerability of SCW positions (e.g. employment on zero-hours contracts). This is consistent with previous research that shows promoting a positive choice around the seasonal influenza vaccine rather than resorting to a more coercive approach, such as mandating, was supportive of fostering continuous improvement of vaccination uptake [24, 27].

Interviews and free-text data brought out powerfully that those who declined the vaccine, or were unsure about receiving it, were calling for "conversations" where they felt safe to ask about the vaccine, and not feel judged and stigmatised for having questions and/or concerns. This is consistent with recent guidance issued by the NHS [28] that recommends '*supportive and sensitive one to one conversation*' with members of staff. Some participants argued that, as educated HCWs, they felt disempowered and should be able to easily access and examine the evidence about the vaccines and scientific information. Ethnic minority participants suggested tailoring vaccination communication to promote uptake amongst H&SCWs from ethnic minority backgrounds, and engaging faith leaders and trusted figures who understand community member concerns. Key messages and policy recommendations are provided in Box 1.

---

### Box 1. Key messages and policy recommendations

#### Strategy

- Build on lessons learned from the seasonal influenza vaccination programme for health and social care workers.

- Respond to information needs rather than focus on the need to vaccinate. Provide a safe space for asking questions and discussing concerns.

- Provide easy access to evidence, and acknowledge that evidence changes over time.

- Promote a positive workplace environment in which vaccination-decisions are informed and voluntary.

- Carefully consider any change in the current policy of voluntary vaccination, and the possible unintended consequences of moving to mandatory vaccination on trust and uptake.

## Decliners

- Do not pressure decliners and/or set up a system that is repeatedly asking them to vaccinate. Provide space for discussing concerns.

- Be careful how vaccination status is recorded in individual staff health records.

- Do not frame specific ethnic or professional groups as "refusers". Do not label decliners negatively or stigmatise specific groups.

## Specific demographic groups

- Promote more "representative" champions of specific ethnic communities; involve clinical leaders; pilot peer to peer discussion.

- Reach out to staff with health conditions; provide relevant information and referrals. Provide regular updates on changes in scientific guidance (e.g. pregnancy).

## Access

- Provide information on fail-safe referral pathways for those not supported by their employers (e.g. social care workers, those with multiple employers).

## Side effects

- Communicate transparently on side effects, respond to questions, and provide further scientific evidence as it emerges.

### Strengths and limitations

This is the first study to consider the role of beliefs, attitudes and other factors in COVID-19 vaccination uptake among H&SCWs. By using a mixed-methods approach, our study explored not only the association between demographic characteristics–as other studies have done [29] —but also attitudinal, organisational and cultural influences on COVID-19 vaccination that have not been examined in UK H&SCWs.

Our recruitment strategy achieved a high number of H&SCW responses from across a range of job roles, and intentionally high representation across broad ethnic categories. On reflection, we consider that this was possible due to diverse representation among the study researchers.

Future research should concentrate on exploring the heterogeneity within broad ethnic groups. While ethnic minority representation was good across our sample, some categories used within the regression analysis still had less than ideal numbers of participants. For instance, there were 66 Black or Black British Caribbean or Mixed Black Caribbean participants, 15 of whom declined the COVID-19 vaccine. With such numbers, it is possible that recruitment bias and other general limitations to conducting survey research may limit the representativeness of the findings. In our sample, Asian participants were overrepresented within highly medical roles (54% in our sample vs 26% in the NHS workforce [20]), suggesting that our higher uptake findings for Asian participants should be interpreted cautiously.

## Conclusion

Our study has provided a nuanced analysis of factors influencing COVID-19 vaccination uptake amongst H&SCWs, exploring variations found in vaccination offer and uptake by job sector and ethnicity. To increase COVID-19 vaccine confidence requires the use of targeted communication and engagement strategies for specific communities that respond to their concerns; reframing the communication strategy to prioritise allowing for and responding to individual questions and concerns; and not stigmatising specific ethnic or professional groups. The role of employers is central to strengthening the H&SCW COVID-19 vaccination programme, given that employers are the natural conduit for providing COVID-19 vaccination information and facilitating vaccine discussions. As COVID-19 vaccination may become a routine vaccination for all HSCWs in the future, it is important to engage H&SCWs in supporting the programme and create a positive and supportive environment for COVID-19 vaccination. Crucially, our findings emphasise the importance of COVID-19 vaccination remaining voluntary, as a move towards mandating COVID-19 vaccination is likely to harden stances and negatively affect trust in the vaccination, provider, and policymakers.

## Supporting information

**S1 Appendix. Survey questions.**
(DOCX)

**S2 Appendix. Recoding of variables.**
(DOCX)

## Acknowledgments

We would like to thank everyone who helped share the online survey with potential participants. We are especially grateful for the time and contribution of all health and social care workers who took part in the study.

We would also like to thank Professor Bob Erens at LSHTM for his support in using Qualtrics software for the survey.

## Author Contributions

**Conceptualization:** Sadie Bell, Richard M. Clarke, Habib Naqvi, Yvonne Coghill, Helen Donovan, Louise Letley, Pauline Paterson, Sandra Mounier-Jack.

**Data curation:** Sadie Bell, Richard M. Clarke.

**Formal analysis:** Sadie Bell, Richard M. Clarke, Pauline Paterson, Sandra Mounier-Jack.

**Investigation:** Sadie Bell, Richard M. Clarke, Sharif A. Ismail, Oyinkansola Ojo-Aromokudu, Pauline Paterson, Sandra Mounier-Jack.

**Methodology:** Sadie Bell, Richard M. Clarke, Habib Naqvi, Yvonne Coghill, Helen Donovan, Louise Letley, Pauline Paterson, Sandra Mounier-Jack.

**Project administration:** Sadie Bell, Sandra Mounier-Jack.

**Supervision:** Sandra Mounier-Jack.

**Visualization:** Sadie Bell, Richard M. Clarke.

**Writing – original draft:** Sadie Bell, Richard M. Clarke.

**Writing – review & editing:** Sadie Bell, Richard M. Clarke, Sharif A. Ismail, Oyinkansola Ojo-Aromokudu, Habib Naqvi, Yvonne Coghill, Helen Donovan, Louise Letley, Pauline Paterson, Sandra Mounier-Jack.

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
