## [Decision Letter · Decision Letter 0]

20 Sep 2021

PONE-D-21-13422COVID-19 vaccination beliefs, attitudes, and behaviours among health and social care workers in the UK: a mixed-methods studyPLOS ONE

Dear Dr. Bell,

Thank you for submitting your manuscript to PLOS ONE. After careful consideration, we feel that it has merit but does not fully meet PLOS ONE’s publication criteria as it currently stands. Therefore, we invite you to submit a revised version of the manuscript that addresses the points raised during the review process.

We look forward to receiving your revised manuscript.

Kind regards,

Maryam Farooqui, Ph.D

Academic Editor

PLOS ONE

Journal Requirements:

6. We note you have included a table to which you do not refer in the text of your manuscript. Please ensure that you refer to Tables 5 and 6 in your text; if accepted, production will need this reference to link the reader to the Table.

Reviewers' comments:

Reviewer's Responses to Questions

**Comments to the Author**

1. Is the manuscript technically sound, and do the data support the conclusions?

Reviewer #1: Yes

Reviewer #2: Yes

2. Has the statistical analysis been performed appropriately and rigorously? 

Reviewer #1: Yes

Reviewer #2: Yes

3. Have the authors made all data underlying the findings in their manuscript fully available?

Reviewer #1: Yes

Reviewer #2: Yes

4. Is the manuscript presented in an intelligible fashion and written in standard English?

Reviewer #1: Yes

Reviewer #2: Yes

5. Review Comments to the Author

Reviewer #1: Overall the manuscript "COVID-19 vaccination beliefs, attitudes and behaviours among health and social care workers in the UK: a mixed methods study" is a well written and meticulously researched study that aims to assess vaccination uptake and attitude in HCW and SCW of different ethnic diversity and occupational groups. The methods used in the study involve both quantitative and a qualitative approach followed by adequately described statistical analysis methods.

If the authors have measured the reliability of the COVID-19 vaccination belief and vaccination information using methods such as the Cronbach's alpha coefficient, it would be useful to include the coefficient in the paper, as this tend to show the consistency of the tools used particularly when perceptions and attitudes of people's environment are assessed.

Overall, the paper should be accepted for publication.

Reviewer #2: Well planned, executed and presented study.

In methodology, each participants received a honorarium of 10 pounds for attending interviews. will it affect the outcome?

how did you validate the questions? any standardization methods used?

6. PLOS authors have the option to publish the peer review history of their article (what does this mean?). If published, this will include your full peer review and any attached files.

Reviewer #1: **Yes: **Nonhlanhla Tlotleng

Reviewer #2: No

---

## [Author Response · Author response to Decision Letter 0]

8 Oct 2021

Academic Editor’s comments:

We have followed PLOS ONE’s style requirements, including those for file naming.

We have reviewed our reference list.

Please can we include the following ‘Funding Information’: 

‘The author(s) disclosed receipt of the following financial support for the research, authorship, and/or publication of this article: The research was funded by the National Institute for Health Research Health Protection Research Unit (NIHR HPRU) in Immunisation at the London School of Hygiene and Tropical Medicine (LSHTM) in partnership with Public Health England (PHE) (reference number NIHR200929); and by the NHS Race and Health Observatory. The views expressed are those of the author(s) and not necessarily those of the NHS, the NIHR, the Department of Health or Public Health England, or the NHS Race and Health Observatory.

SAI is supported by a Wellcome Trust Clinical Research Training Fellowship [reference number 215654/Z/19/Z].’

4. We note that you have indicated that data from this study are available upon request. PLOS only allows data to be available upon request if there are legal or ethical restrictions on sharing data publicly. For more information on unacceptable data access restrictions, please see http://journals.plos.org/plosone/s/data-availability#loc-unacceptable-data-access-restrictions

• If there are ethical or legal restrictions on sharing a de-identified data set, please explain them in detail (e.g., data contain potentially sensitive information, data are owned by a third-party organization, etc.) and who has imposed them (e.g., an ethics committee). Please also provide contact information for a data access committee, ethics committee, or other institutional body to which data requests may be sent. If there are no restrictions, please upload the minimal anonymized data set necessary to replicate your study findings as either Supporting Information files or to a stable, public repository and provide us with the relevant URLs, DOIs, or accession numbers. For a list of acceptable repositories, please see http://journals.plos.org/plosone/s/data-availability#loc-recommended-repositories.

Here is the amended data availability statement: 

‘The full quantitative survey dataset is stored in a data repository: LSHTM Data compass. Data are available to bona fide researchers upon request and agreement by the study team. Data cannot be shared without restriction as study participants agreed that their study data could be accessed by other researchers only. This level of data access was approved by the LSTHM Observational Research Ethics committee (study reference: 22923). Although permission was granted by participants to share their interview and open-text survey data, there are concerns about potential harms that could come to participants given the content of this data. Therefore, we are sharing the anonymised quantitative survey data only.

Further information on the data and access conditions can be found through the LSHTM Data Compass at: https://doi.org/10.17037/DATA.00002525 . Researchers interested in accessing the data are advised to request access through the LSHTM Data Compass page listed above, email the corresponding author or email researchdatamanagement@lshtm.ac.uk.’

The ethics statement only appears in the Methods section of our manuscript. 

6. We note you have included a table to which you do not refer in the text of your manuscript. Please ensure that you refer to Tables 5 and 6 in your text; if accepted, production will need this reference to link the reader to the Table.

Tables 5 and 6 are now referred to on pages 21 and 23.

We have included the following captions for our Supporting Information files (page 50):

S1 Appendix: Survey questions

S2 Appendix: Recoding of variables

Reviewers' comments:

Reviewer's Responses to Questions

1. Is the manuscript technically sound, and do the data support the conclusions?

Reviewer #1: Yes

Reviewer #2: Yes

2. Has the statistical analysis been performed appropriately and rigorously? 

Reviewer #1: Yes

Reviewer #2: Yes

3. Have the authors made all data underlying the findings in their manuscript fully available?

Reviewer #1: Yes

Reviewer #2: Yes

4. Is the manuscript presented in an intelligible fashion and written in standard English?

Reviewer #1: Yes

Reviewer #2: Yes

Reviewers’ Comments to the Author

Reviewer #1: Overall the manuscript "COVID-19 vaccination beliefs, attitudes and behaviours among health and social care workers in the UK: a mixed methods study" is a well written and meticulously researched study that aims to assess vaccination uptake and attitude in HCW and SCW of different ethnic diversity and occupational groups. The methods used in the study involve both quantitative and a qualitative approach followed by adequately described statistical analysis methods.

If the authors have measured the reliability of the COVID-19 vaccination belief and vaccination information using methods such as the Cronbach's alpha coefficient, it would be useful to include the coefficient in the paper, as this tend to show the consistency of the tools used particularly when perceptions and attitudes of people's environment are assessed.

Overall, the paper should be accepted for publication.

Response to reviewer #1: An extended explanation, including the Cronbach’s alpha coefficient, has now been added for each of the combined variables. This can be seen in manuscript section 2.1.2 and S2 Appendix which presents the full factor analysis used to aid the item combination.

Changes to manuscript (lines 143-149): In addition, we performed a factor analysis on the Vaccine belief and Trust in information source items to reduce the number of variables in the regression models. This reduced the 13 Vaccine belief items into two composite variables; Combined COVID-19 Vaccine belief (important, safe, and effective) (Cronbach’s Alpha = .918) and Social norms to vaccinate against COVID-19 (Cronbach’s Alpha = .661) and 4 single items. The 12 Trust in information source items were reduced to three composite variables Trust in health system sources (Cronbach’s Alpha = .876), Trust in non-health system sources (Cronbach’s Alpha = .738), and Trust in Friends and Family members (Cronbach’s Alpha = .876). Further details related to the combining of the ethnicity and job role categories and the factor analysis can be found in S2 Appendix. Appendix 2. 

Reviewer #2: Well planned, executed and presented study.

In methodology, each participants received a honorarium of 10 pounds for attending interviews. will it affect the outcome?

how did you validate the questions? any standardization methods used?

Authors’ response to reviewer #2: 

Thank you to reviewer #2 for taking the time to read and review our manuscript. We really appreciate your feedback. In response to your questions:

• Interview participants only – not survey respondents - received a £10 gift voucher as a thank you for their time and contribution to the research. We considered the value of the gift voucher appropriate to acknowledge the time and contribution of interview participants, and that the value did not present undue influence. The £10 gift voucher for interview participants was approved by the London School of Hygiene & Tropical Medicine (LSTHM) Observational Research Ethics committee (study reference: 22923). 

• Several of the questions included in the survey were pre-existing survey questions that had already undergone testing (e.g. questions included in the demographics section). New questions were developed and discussed with the study collaborators from Public Health England, the Royal College of Nursing, and the NHS Race and Health Observatory. The survey was also pre-tested with a number of health and social care workers to explore the appropriateness and comprehensibility of the survey questions. Additional detail in response to this question has now been added to section 2.1.1. 

• No standardization methods were used for the data cleaning/analysis.

Changes to manuscript (lines 124-132): 

• 'We developed the survey in consultation with Public Health England, Royal College of Nursing and NHS Race and Health Observatory representatives. Several included questions were pre-existing survey questions that had already undergone testing (e.g. questions included in the demographics section). For new items and questions, face validity was gained through discussion with the various research stakeholders, and feedback on survey design (including the appropriateness and comprehensibility of questions) was obtained from a number of H&SCWs. Factor analysis (see S2 Appendix) identified and confirmed underlying components of the Vaccine belief and Trust in information source items. Public Health England, Royal College of Nursing and NHS Race and Health Observatory representatives were also involved in sharing the survey. Feedback on survey design was also obtained from a number of H&S

---

## [Editor Report · Decision Letter 1]

22 Nov 2021

COVID-19 vaccination beliefs, attitudes, and behaviours among health and social care workers in the UK: a mixed-methods study

PONE-D-21-13422R1

Dear Dr. Bell,

We’re pleased to inform you that your manuscript has been judged scientifically suitable for publication and will be formally accepted for publication once it meets all outstanding technical requirements.

Kind regards,

Maryam Farooqui, Ph.D

Academic Editor

PLOS ONE
---

## [Editor Report · Acceptance letter]

9 Dec 2021

PONE-D-21-13422R1 

COVID-19 vaccination beliefs, attitudes, and behaviours among health and social care workers in the UK: a mixed-methods study 

Dear Dr. Bell:

I'm pleased to inform you that your manuscript has been deemed suitable for publication in PLOS ONE. Congratulations! Your manuscript is now with our production department. 

Kind regards, 

on behalf of

Dr. Maryam Farooqui 

Academic Editor

PLOS ONE